# DIFFUSION-BASED SYMBOLIC REGRESSION

## ABSTRACT

Diffusion has emerged as a powerful framework for generative modeling, achieving remarkable success in applications such as image and audio synthesis. Enlightened by this progress, we propose a novel diffusion-based approach for symbolic regression. We construct a random mask-based diffusion and denoising process to generate diverse and high-quality equations. We integrate this generative processes with a token-wise Group Relative Policy Optimization (GRPO) method to conduct efficient reinforcement learning on the given measurement dataset. In addition, we introduce a long short-term risk-seeking policy to expand the pool of top-performing candidates, further enhancing performance. Extensive experiments and ablation studies have demonstrated the effectiveness of our approach.

## 1 Introduction

Given a dataset of measurements $\mathcal{D} = \{(\mathbf{x}_i, y_i)\}_{i=1}^{N}$, symbolic regression aims to discover a simple mathematical expression that captures the relationship between the input and output variables, such as $y = 3\sin(x_1) + x_2^2$. Unlike traditional machine learning, where the model architecture is fixed, symbolic regression explores an open-ended space, dynamically adjusting the number, order, and type of parameters and operations. While machine learning models can also be written as mathematical expressions, they are often too complicated or opaque in form for humans to understand. Symbolic regression prioritizes simplicity and interpretability, making it especially popular among scientists and engineers who seek not only for accurate predictions but also a deeper understanding of the underlying data relationships. Interpretable models also earn greater trust, as they avoid unexplained behaviors and require less extensive testing for validation. In contrast, large, complex models often behave unpredictably, especially in regions with sparse training data.

Since its publication in 1994, genetic programming (GP) (Koza, 1994; Randall et al., 2022; Burlacu et al., 2020) has been the dominant approach to symbolic regression. It begins with a population of randomly generated seed expressions and iteratively evolves the population through genetic operations such as selection, crossover, and mutation, until a set of optimal equations is found. Despite its strong performance, GP is known to be computationally expensive due to the need for many generations and extensive genetic operations. To address this, Petersen et al. (2019) proposed Deep Symbolic Regression (DSR), which significantly accelerates expression discovery. DSR introduces a recurrent neural network (RNN) to sample expressions and employs a reinforcement learning framework, with a risk-seeking policy gradient, to train the RNN on the measurement dataset. DSR has since become a major baseline in symbolic regression research and development. More recent efforts have explored using pretrained foundation models (Kamienny et al., 2022; Valipour et al., 2021) to map datasets directly to candidate expressions, followed by GP and/or Monte Carlo Tree Search (MCTS) (Browne et al., 2012) to further optimize the expression(s) for a given dataset.

Most of the recent symbolic regression (SR) approaches rely on a generative model for expression sampling, trained by maximizing the likelihood of the correct next token. However, diffusion methods — as a powerful generative modeling framework (Ho et al., 2020) — have been relatively overlooked in SR, despite their remarkable success in other domains, such as image generation (Rombach et al., 2022), audio synthesis (Huang et al., 2023), and more recently, large language model (LLM) training (Nie et al., 2025). Diffusion models apply a forward process that gradually corrupts data with noise, while learning a reverse process that denoises to reconstruct the original instances. New samples are then generated by starting from random noise and iteratively applying the reverse denoising steps. This mechanism enables diffusion models to produce more diverse and high-quality samples, offering the potential to improve SR by exploring expression space more

effectively and avoiding collapse into a single inferior mode (*e.g.*, overly complex structures or spurious terms), thereby facilitating the discovery of better expressions. Motivated by this, we propose a diffusion-based deep symbolic regression method (DDSR) for generating expressions from measurement datasets. Our main contributions are summarized as follows.

- **Random Mask-Based Discrete Diffusion**. We propose a discrete diffusion model for expression generation, where noise is represented by token masking. The forward process randomly masks out one token at a time. Generation starts with a fully masked (empty) sequence and progressively reconstructs the tokens step by step. This approach not only enables the generation of diverse expressions but also significantly reduces the number of denoising steps and the overall computational cost.
- **Token-Wise GRPO**. We integrated our diffusion model into a Group Relative Policy Optimization (GPRO) (Shao et al., 2024a) framework for efficient reinforcement learning. At each step, we employ a risk-seeking strategy by selecting the top-performing expressions generated by our model. We maximize the per-token denoising likelihood for each expression, scaled by its corresponding reward. The GRPO framework enforces updates within a trust region, thereby improving both the stability and efficiency of the learning process.
- **Long Short-Term Risk-Seeking**. We extend the risk-seeking policy used in DSR, which selects top-performing expressions solely from the current model. While effective locally, this strategy may focus too much on short-term improvements and overlook longer-term trends. To address this, we expand the candidate pool to include top-performing expressions sampled from all model versions seen so far. This combined strategy resolves both long-term and short-term risks, aiming to build a more robust and effective model.
- **Experiments.** We evaluated DDSR on the SRBench benchmark, comparing it against eighteen baseline methods. Our results show that DDSR significantly improves both solution accuracy and symbolic recovery rate on datasets with known ground-truth expressions, as compared to DSR. Moreover, DDSR achieves a higher symbolic solution rate than most genetic programming (GP) methods, while generating considerably simpler and more interpretable expressions. On the black-box problems, DDSR lies on the Pareto frontier, demonstrating a favorable trade-off between expression complexity and predictive performance. Ablation studies further validate the contribution of each individual component in our framework, confirming their collective importance to overall performance.

## 2 Background

**Diffusion Models.** The denoising diffusion probabilistic model (DDPM) (Ho et al., 2020) fundamentally shifted the paradigm of generative modeling and has inspired numerous follow-up works. However, DDPM defines the diffusion process as a Gauss-Markov chain that gradually adds Gaussian white noise, and is therefore inherently suited for continuous data. For categorical data such as discrete tokens, adding continuous Gaussian noise is neither feasible nor meaningful. To address this issue, Austin et al. (2021) proposed the Discrete Denoising Diffusion Probabilistic Model (D3PM). Each data instance $\mathbf{X}_0 \in \mathbb{R}^{M \times d}$ represents a collection of $M$ tokens, where each row is the one-hot encoding of a token (assuming $d$ different categories for each token). D3PM defines a forward process that gradually transforms the deterministic one-hot encoding $\mathbf{X}_0$ into a uniform distribution, effectively modeling *discrete* white noise. Specifically, at each step $t > 0$, the token distribution is updated via $\mathbf{X}_t = \mathbf{X}_{t-1}\mathbf{Q}_t$, where $\mathbf{Q}_t = \beta_t \mathbf{I} + (1 - \beta_t)\mathbf{1}\mathbf{1}^\top / d$, $\mathbf{1}$ is a vector of ones, and $\beta_t \in (0, 1)$. It can be shown that each row of $\mathbf{X}_t$ maintains a valid probability distribution, and as $t \to \infty$, each row converges to the uniform distribution. One can derive a closed-form conditional distribution for sampling: $q(\mathbf{X}_t|\mathbf{X}_0) = \mathbf{X}_0 \overline{\mathbf{Q}}_t$ where $\overline{\mathbf{Q}}_t = \mathbf{Q}_1 \mathbf{Q}_2 \cdots \mathbf{Q}_t$, and $q(\mathbf{X}_{t-1}|\mathbf{X}_t, \mathbf{X}_0) = \frac{\mathbf{X}_t \mathbf{Q}_t^\top \odot \mathbf{X}_0 \overline{\mathbf{Q}}_{t-1}}{\mathbf{X}_0 \overline{\mathbf{Q}}_t \mathbf{X}_t^\top}$, where $\odot$ is element-wise multiplication. During training, a random timestep $t$ is selected, and tokens are sampled from $q(\mathbf{X}_t|\mathbf{X}_0)$. The sampled tokens, along with $t$, are fed into a neural network tasked with predicting the initial token distribution $q(\mathbf{X}_0)$. The model is trained by minimizing a cross-entropy loss between the predicted distribution and the ground-truth tokens.

Generation starts with randomly sampled tokens from the uniform distribution. At each step $t$, the conditional distribution $q(\mathbf{X}_{t-1}|\mathbf{X}_t)$ is computed by marginalizing out $\mathbf{X}_0$ in $q(\mathbf{X}_{t-1}|\mathbf{X}_t, \mathbf{X}_0)$ with the distribution $q(\mathbf{X}_0)$ predicted by the neural network. A sample $\mathbf{X}_{t-1}$ is drawn accordingly. This process repeats until $t = 0$, at which $\mathbf{X}_0$ is obtained as the final generated sample.

**Deep Symbolic Regression (DSR).** Given a measurement dataset, DSR trains a recurrent neural network (RNN) to generate expressions that describe the underlying data. The RNN predicts each token in the preorder traversal of the expression tree in an autoregressive manner. Training is performed via reinforcement learning, where the reward is based on the normalized root mean squared error (NRMSE) of the data fit: $\mathrm{NRMSE} = \frac{1}{\sigma_y}\sqrt{\frac{1}{n}\sum_{i=1}^{n}(y_i - \tau(\mathbf{x}_i))^2}$, where $\tau$ denotes the expression and $\sigma_y$ is the standard deviation of the outputs in the dataset. Since the goal is to identify the best expressions, DSR employs a risk-seeking policy in which only the top $\alpha\%$ of expressions are used to update the model at each iteration:

$$R(\tau) = \frac{1}{1 + \mathrm{NRMSE}(\tau, \mathbf{x}, y)}, \tag{1}$$

$$\nabla J_{\mathrm{risk}}(\theta; \alpha) = \frac{1}{B\alpha/100}\sum_{i=1}^{B}[R(\tau^{(i)}) - R_\alpha] \cdot \mathbb{1}\left(R(\tau^{(i)}) \geq R_\alpha\right)\nabla_\theta \log(p(\tau^{(i)}|\theta)), \tag{2}$$

where $B$ is the size of expression batch sampled at each epoch, $R_\alpha$ is the minimum reward among the top $\alpha\%$ expressions, $R(\tau)$ is the reward for any expression $\tau$, $\mathbb{1}(\cdot)$ is an indicator function, and $\theta$ denotes the parameters of the RNN.

## 3  Method

### 3.1  Random Mask-Based Discrete Diffusion

We represent a symbolic expression as a token matrix $\mathbf{X}_0 \in \mathbb{R}^{M \times d}$, where each row is the one-hot encoding of a token and $M$ denotes the maximum number of tokens. If the actual number of tokens is fewer than $M$, we pad the matrix with zero rows. While one could directly apply the D3PM method for expression generation (see Section 2), we empirically found its performance to be unsatisfactory. In D3PM, at each diffusion and denoising step, the distribution of every token is perturbed, which can severely disrupt the structure of the expression. This disruption leads to unstable and inefficient training, particularly when combined with reinforcement learning, resulting in degraded performance.

Recent work of Nie et al. (2025) on large language models proposed randomly masking a portion of sequence elements at each step and training the model to reconstruct the masked elements conditioned on the remaining ones. Inspired by their success, we adopt a similar idea but with a key difference: *we mask out only one token at each step*. This approach gradually and smoothly blurs the expression structure, avoiding abrupt distortions and preserving most structural information, thereby promoting learning stability and efficiency.

Specifically, let $q_t$ denote the token index to be masked at time step $t$, and let $\overline{q}_t = \{q_1, \ldots, q_t\}$ represent the set of all masked indices up to step $t$. Given $\mathbf{X}_0$, we sample $q_t$ and $\mathbf{X}_t$ as follows:

$$q_t \sim \mathrm{Uniform}(\{1, \ldots, M\}\backslash\overline{q}_{t-1}), \qquad \mathbf{Q}_t = \mathbf{I} - \mathrm{diag}(\mathbf{e}_{q_t}),$$

$$\mathbf{X}_t = \overline{\mathbf{Q}}_t\mathbf{X}_0, \qquad \overline{\mathbf{Q}}_t = \mathbf{Q}_t\mathbf{Q}_{t-1}\ldots\mathbf{Q}_1, \tag{3}$$

where $\mathbf{e}_{q_t}$ is a one-hot vector with one at position $q_t$ and zeros elsewhere. We design a Transformer network $\phi_\theta$ that takes $\mathbf{X}_t$ as input and predicts $q(\mathbf{X}_0)$ — the token distribution matrix corresponding to $\mathbf{X}_0$. The architectural details of $\phi_\theta$ are provided in Appendix B. Training of $\phi_\theta$ is integrated into a reinforcement learning framework, whose details are described later.

To generate an expression, we begin with a zero matrix $\mathbf{X}_M$, where all tokens are masked. At each backward step $t = M, M-1, \ldots$, we input $\mathbf{X}_t$ into $\phi_\theta$ to predict the distribution $q(\mathbf{X}_0)$ and use it to sample the masked tokens in $\mathbf{X}_t$. To ensure validity, we first identify the valid token set for each masked position based on the current unmasked tokens. We retrieve and normalize the probabilities of these valid tokens from $q(\mathbf{X}_0)$, and sample each masked token accordingly. The sampled tokens are combined with the existing unmasked tokens to form an intermediate instance of $\mathbf{X}_0$. Next, we apply the diffusion (masking) process defined in (3) to obtain $\mathbf{X}_{t-1}$, while keeping all the previously unmasked tokens fixed, *i.e.,* revealing exactly one new token.

This iterative process continues, reconstructing one token at a time, until a complete sample of $\mathbf{X}_0$ is obtained (*i.e.,* $t = 0$). Generation terminates early if the current token matrix already forms a valid expression. If the final $\mathbf{X}_0$ does not represent a valid expression, we randomly replace invalid tokens until a valid expression is obtained. The full generation procedure is summarized in Appendix Algorithm 2, and further implementation details are given in Appendix Section A.1.

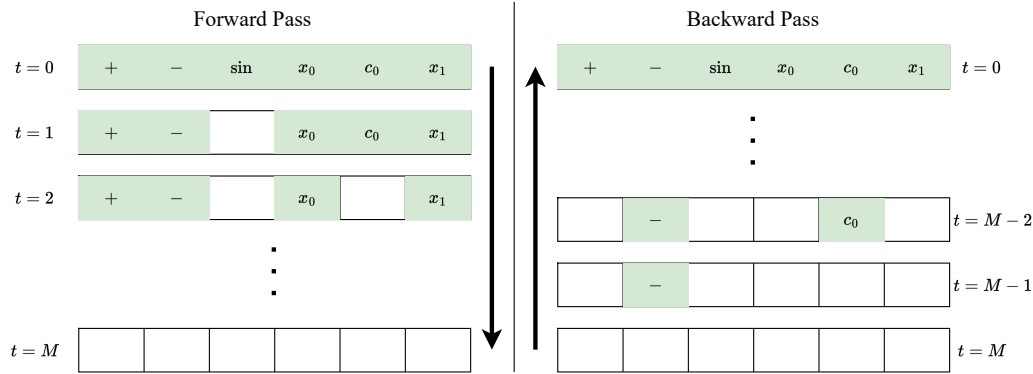

Figure 1: Illustration of the forward and backward process of the random mask-based diffusion. White entries represent the masked tokens. Green entries represent tokens in the original expression (left), and the generated tokens (right).

## 3.2 Reinforcement Learning with Token-Wise GPRO

To train the diffusion model $\phi_\theta$ using a given measurement dataset $\mathcal{D}$, we adopt a reinforcement learning framework. Specifically, we employ a risk-seeking strategy similar to that used in DSR. At each training step, we select the top $\alpha\%$ of expressions generated by our model based on their NRMSE of the data fit. For each expression $\tau^{(i)}$, we assign a reward $R(\tau^{(i)})$ as defined in (1). We then update $\phi_\theta$ to encourage the generation of expressions with similarly high rewards.

Specifically, let $\mathbf{X}_0^{(i)}$ denote the token matrix representation of expression $\tau^{(i)}$. We randomly select a diffusion step $t$ and generate a noisy version $\mathbf{X}_t^{(i)}$ using the forward process as described in (3). Feeding $\mathbf{X}_t^{(i)}$ into $\phi_\theta$, we obtain a prediction of $q(\mathbf{X}_0^{(i)})$. The model is trained to maximize the log-likelihood of $\mathbf{X}_0^{(i)}$ scaled by the relative reward $A_i = R(\tau^{(i)}) - R_\alpha$, under the predicted $q(\mathbf{X}_0^{(i)})$. This leads to the following optimization objective:

$$\text{maximize}_\theta \quad \mathbb{E}_t \left[ \sum_{\tau^{(i)} \in \mathcal{S}_\alpha} A_i \cdot \log p \left( \mathbf{X}_0^{(i)} | \phi_\theta(\mathbf{X}_t^{(i)}) \right) \right], \tag{4}$$

where $\mathcal{S}_\alpha$ denotes the set of top $\alpha\%$ expressions, and the likelihood $p(\mathbf{X}_0^{(i)}|\phi_\theta(\mathbf{X}_t^{(i)})) = \prod_k p(\mathbf{x}_k^{(i)}|\boldsymbol{\eta}_{k,t,\theta}^{(i)})$, each $\mathbf{x}_k^{(i)}$ is the $k$-th row of $\mathbf{X}^{(i)}$ corresponding to an non-empty token, and $\boldsymbol{\eta}_{k,t,\theta}^{(i)}$ is the predicted probability distribution for that token, *i.e.,* the $k$-th row of $\phi_\theta(\mathbf{X}_t^{(i)})$.

To further enhance training stability and efficiency, we adapt the recent Group Relative Policy Optimization (GRPO) framework (Shao et al., 2024b) to optimize (4). Specifically, instead of using all token likelihoods equally, we selectively use them to update model parameters based on a trust region criterion: we only use token likelihoods whose variation remains within a bounded range relative to the earlier model. Specifically, the model update is given by:

$$\theta \leftarrow \theta + \gamma \nabla J_{\text{GRPO}}(\theta; \alpha),$$

$$J_{\text{GRPO}}(\theta; \alpha) = \frac{1}{B\alpha/100} \sum_{\tau^{(i)} \in \mathcal{S}_\alpha} \sum_k \min \left\{ h_{\theta k t}, \text{clip}\left(h_{\theta k t}, 1 - \epsilon, 1 + \epsilon\right) \right\} \cdot A_i - \beta \cdot g_{\theta k t}, \tag{5}$$

where $\gamma > 0$ is the learning rate, $h_{\theta k t} = \frac{p(\mathbf{x}_k^{(i)}|\boldsymbol{\eta}_{k,t,\theta}^{(i)})}{p(\mathbf{x}_k^{(i)}|\boldsymbol{\eta}_{k,t,\theta_{\text{old}}}^{(i)})}$ is the likelihood ratio of $k$-th token between the current model and the model at the beginning of the current epoch (with parameters $\theta_{\text{old}}$), $g_{\theta k t} = \text{KL}[p(\mathbf{x}_k^{(i)}|\boldsymbol{\eta}_{k,t,\theta}^{(i)}) \| p(\mathbf{x}_k^{(i)}|\boldsymbol{\eta}_{k,t,\theta_{\text{ref}}}^{(i)})]$ is a regularization term that regularizes the current model not to deviate too much from a reference model with parameters $\theta_{\text{ref}}$. Here $\beta > 0$ controls the strength of regularization, and $\text{KL}[\cdot\|\cdot]$ denotes the Kullback–Leibler divergence. By restricting updates to tokens whose likelihood ratios $h_{\theta k t}$ lie within the trust region $[1 - \epsilon, 1 + \epsilon]$, we prevent unstable and potentially harmful updates caused by outlier token samples.

It is worth noting that our approach differs from standard diffusion training, which constructs fixed paths from ground-truth instances to noise and learns to reverse them. Our setting is *unsupervised*:

---

**Algorithm 1** Diffusion based Deep Symbolic Regression (DDSR)

---

**input** Learning rate $\gamma$; risk factor $\alpha$; expression batch size $B$; steps per epoch $C$; epochs per reference $G$; number of epochs $N$; entropy scalar $\lambda$

**output** The best equation $\tau^*$

  1: Initialize transformer $\phi$ with parameters $\theta$
  2: $\mathcal{S}_\alpha \leftarrow \{\}$
  3: **for** $i = 0$ to $N - 1$ **do**
  4:     **if** $i \bmod(G) = 0$ **then**
  5:         $\theta_{\text{ref}} \leftarrow \theta$
  6:     **end if**
  7:     $\theta_{\text{old}} \leftarrow \theta$
  8:     Sample $B$ expression with the current model, and obtain the top $\alpha\%$ expressions $\mathcal{S}_\alpha^i$
  9:     $\mathcal{S}_\alpha \leftarrow \mathcal{S}_\alpha \cup \mathcal{S}_\alpha^i$
10:     Set $R_\alpha$ to the minimum award among the expressions in $\mathcal{S}_\alpha$.
11:     **for** $j = 1$ to $C$ **do**
12:         Randomly sample diffusion time step $t$
13:         Compute $J_{\text{GRPO}}$ from (5)
14:         $\theta \leftarrow \theta + \gamma(\nabla_\theta J_{\text{GRPO}} + \lambda \cdot \text{Entropy-Gradient})$
15:     **end for**
16:     Remove the bottom $\alpha\%$ expressions from $\mathcal{S}_\alpha$ according to the rewards
17: **end for**
18: **return** the best expression from $\mathcal{S}_\alpha$

---

we only have numerical measurement data, with no ground-truth expressions available (the goal is to discover them). To address this, we adopt a reinforcement learning approach. Starting from randomly initialized parameters, the diffusion model generates candidate expressions, which are evaluated on the measurement data and assigned rewards. These rewards weight the log-likelihoods of the corresponding token sequences, forming the update in (5). Since token likelihoods are determined by the diffusion model, gradients can be backpropagated to update parameters, progressively increasing the probability of generating high-reward expressions. Repeating this process throughout training adaptively shifts the model distribution toward more accurate expressions.

### 3.3 Long Short-Term Risk-Seeking

The risk-seeking policy used in DSR leverages only the top-performing expressions generated by the current model. However, this local policy can limit the exploitation capability of reinforcement learning. To encourage broader exploitation and to drive the model toward generating diverse yet high-quality expressions, we expand the candidate set by incorporating the top $\alpha\%$ expressions not only from the current epoch but also from all previous epochs. Specifically, at each epoch $k$, we update the candidate pool as follows:

$$\mathcal{S}_\alpha \leftarrow \mathcal{S}_\alpha \cup \mathcal{S}_\alpha^k, \tag{6}$$

where $\mathcal{S}_\alpha^k$ denotes the top $\alpha\%$ expressions sampled at epoch $k$. We then set $R_\alpha$ to the minimum reward among all expressions in $\mathcal{S}_\alpha$, and update the model accordingly following (5). To prevent the buffer from growing indefinitely, after each epoch we remove the bottom $\alpha\%$ expressions with the lowest rewards from $\mathcal{S}_\alpha$. This policy can be viewed as a hybrid of the risk-seeking strategy in DSR and the priority training queue proposed in (Mundhenk et al., 2021). By jointly considering both short-term and long-term risks, our approach prevents the model from drifting away from well-performing but hard-to-sample expressions. By continuously updating from such expressions, the model progressively improves its ability to generate high-quality outputs. A full summary of our training procedure is provided in Algorithm 1.

## 4 Related Work

The Deep Symbolic Regression (DSR) framework (Petersen et al., 2019) pioneered the use of reinforcement learning to train RNN-based expression generators from the measurement data. Building on this framework, recent extensions (Tenachi et al., 2023; Jiang et al., 2024) have enforced physics-unit constraints as domain knowledge to enhance the quality of expression generation or confined the search space through vertical discovery strategies for vector symbolic regression.

Another line of work has shifted toward building foundation models that map numerical measurements outright to symbolic expressions (Biggio et al., 2021; Kamienny et al., 2022; Valipour et al., 2021; Vastl et al., 2022). These models typically adopt an encoder-decoder transformer architecture, where encoder layers extract structural patterns from the input data, and decoder layers synthesize symbolic outputs. Although these approaches are promising, the training is costly and acutely sensitive to data preparation pipelines, often requiring massive synthetic datasets. More critically, without a data-specific search mechanism, such models often struggle to generalize — especially when faced with out-of-distribution measurement datasets (Kamienny et al., 2023).

To overcome these limitations, a new wave of research couples pretrained foundation models with explicit search or planning mechanisms tailored to the target dataset, for instance, TPSR (Shojaee et al., 2023), DGSR (Holt et al., 2023), and DGSR-MCTS (Kamienny et al., 2023). In TPSR, the pretrained model is integrated into a modified Monte Carlo Tree Search (MCTS) method (Browne et al., 2012), using model-guided token selection and tree expansion driven by an upper confidence bound (UCB) heuristic. DGSR combines a pretrained encoder-decoder model with genetic programming (GP) at inference time: decoder-generated expressions seed the initial GP population, and top candidates are used to iteratively fine-tune the decoder. DGSR-MCTS uses a flexible MCTS search model including a mutation policy network, augmented with critic layers; The mutation policy network is pretrained on external datasets and then fine-tuned on the task-specific data while the critic layers are trained from scratch on the task-specific data.

Beyond these model-based approaches, ensemble frameworks such as uDSR (Landajuela et al., 2022) combine multiple symbolic regression strategies (*e.g.*, GP and DSR) to boost robustness and accuracy. Meanwhile, model-free methods (Sun et al., 2023; Xu et al., 2024) explore purely search-based techniques, using MCTS or ensemble strategies combining MCTS with GP, to uncover symbolic expressions without relying on heavy pretraining.

In the domain of discrete diffusion, beyond D3PM (Austin et al., 2021), several works have explored graph (Vignac et al., 2023) and tree (Li et al., 2024) generation. Both graph diffusion models employ a modified transition matrix of the form $\mathbf{Q}_t = (1 - \beta_t)\mathbf{I} + \beta_t \mathbf{1m}^\top$, where $\beta_t$ is a scalar determined by the diffusion schedule, and $\mathbf{m}$ represents the marginal distribution over nodes (or edges) in the training data. Recently, Nie et al. (2025) proposed a masked discrete distribution framework, training models in a supervised learning setting to improve sampling quality in LLMs. Our approach adopts a similar masking idea but differs in two key aspects: (1) at each step, we mask out only a single token to preserve structural information more effectively, and (2) we integrate the diffusion process into a reinforcement learning framework to guide training toward high-reward expressions.

## 5 Experiment

### 5.1 Performance on SRBench

We first evaluated DDSR on the well-known and comprehensive SRBench dataset (La Cava et al., 2021), which is divided into two groups: 133 problems with known ground-truth solutions and 120 black-box problems without known solutions. The black-box problems consists of real-world scientific and engineering problems spanning domains such as health informatics, technology, environmental science, and economics. For the first group, four noise levels are considered: 0%, 0.1%, 1%, and 10% (see details in Appendix Section C). All experiments were conducted on A40s from the NCSA Delta cluster[1]. Each A40 ran 8 trials in parallel. For each problem at each noise level, we ran DDSR eight times, with each run capped at a four-hour time limit. The hyperparameter settings used by DDSR are detailed in Appendix Table 4.

We compared DDSR against eighteen existing symbolic regression (SR) methods, spanning GP-based, MCTS-based, deep learning-based, and ensemble approaches. Notably, we include two versions of DSR in the comparison: the original version without constant tokens and an extended version that incorporates and optimizes constant tokens during training. We denote these as DSR-W/OC and DSR-W/C, respectively. *DSR is the most comparable method to DDSR*, as it also uses a reinforcement learning framework and learns directly from data, without relying on pretrained models. AIFeynman is a method that searches for hyperplanes in the dataset and fits each with a polynomial (Udrescu & Tegmark, 2020). Gplearn, Bingo (Randall et al., 2022), GP-GOMEA (Virgolin et al., 2021),

---

[1]https://www.ncsa.illinois.edu/research/project-highlights/delta/

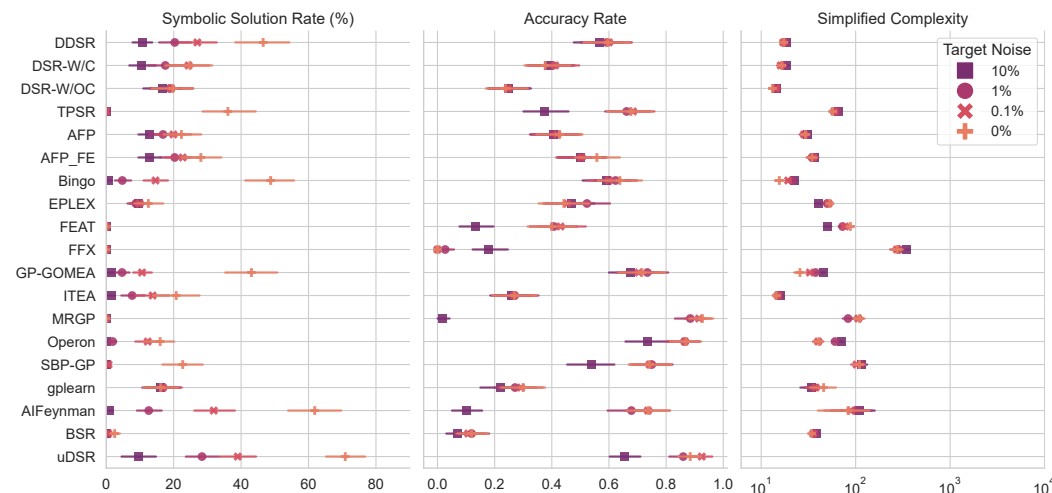

Figure 2: Performance on SRBench problems with ground-truth solutions. Error bars denote a 95% confidence interval. uDSR's high performance is attributed to it being an ensemble method that combines several SR methodologies.

Table 1: Performance on SRBench problems with ground truth solutions.

| Algorithm | Symbolic Solution Rate (%) | | | | Accuracy Rate (%) | | | |
|---|---|---|---|---|---|---|---|---|
| | 0.0 | 0.001 | 0.01 | 0.1 | 0.0 | 0.001 | 0.01 | 0.1 |
| DDSR | **46.54** | **27.02** | **20.33** | 10.69 | 60 | 60 | 59 | 56 |
| DSR-W/C | 24.81 | 24.42 | 17.53 | 10.48 | 38 | 41 | 40 | 39 |
| DSR-W/OC | 19.71 | 19.23 | 18.92 | **16.61** | 24 | 25 | 25 | 25 |
| GP-GOMEA | 43.08 | 10.62 | 4.69 | 1.46 | **71** | **70** | **73** | **68** |
| TPSR | 36.09 | 0.00 | 0.00 | 0.00 | 68 | 68 | 66 | 38 |

SBP-GP (Virgolin et al., 2019), Operon (Burlacu et al., 2020), and MRGP (Arnaldo et al., 2014) all incorporate GP as a primary or supporting component. TPSR combines a pretrained transformer-based foundation model with a Monte Carlo Tree Search to find optimal expressions, while uDSR is a *hybrid* model that ensembles a pretrained numeral symbolic regression model with DSR, GP, AIFeynman, and linear models.

For problems with known solutions, we evaluated each method on symbolic solution rate, accuracy rate, and simplified complexity. The symbolic solution rate is computed using two criteria: symbolic equivalence, as determined by the SymPy library[2], or an $R^2$ score of exactly 1.0. Accuracy rate is a binary metric indicating whether the method finds an expression with $R^2 > 0.999$. The simplified complexity is defined as the number of tokens in the expression tree after simplification by SymPy. We present results for the most comparable and several representative methods in Table 1. Full comparison results are provided in Figure 2 and Appendix Table 5.

We observe that DDSR substantially outperforms DSR in nearly all settings, with the exception of the 10% noise level, where DSR-W/OC achieves a higher solution rate. This may be attributed to the larger token space employed by DDSR. Since DSR-W/OC excludes constant tokens, its reduced token space makes the search process more conservative, potentially offering greater robustness in the presence of high noise. However, even at the 10% noise level, DDSR still achieves substantially higher solution accuracy than DSR-W/OC. When DSR uses the same to-ken space as DDSR— namely, in the DSR-W/C variant — both its solution rate and accuracy consistently lag behind DDSR, demonstrating the advantage of our diffusion-based framework. DDSR also outperforms TPSR, GP-GOMEA, SBP-GP, and many other GP-based methods in terms of solution rate, particularly under non-zero noise levels. Although these methods often achieve higher solution accuracy, the expressions they generate tend to be much *longer* and *more complex*. For instance, TPSR and GP-GOMEA yield average simplified complexities of 61.4 and 35.4,

---

[2]https://www.sympy.org/

respectively, while DDSR maintains a significantly lower average of 17.7. These results underscore the robustness of DDSR and its ability to generate more interpretable symbolic expressions.

AIFeynman outperforms DDSR in symbolic solution rate in the noiseless setting, which aligns with its algorithmic design. AIFeynman fits augmented polynomials and excels when the target expression is a clean polynomial — common among the noiseless symbolic problems in SRBench. However, its performance degrades sharply in black-box settings; its average $R^2$ score falls below zero, leading to its exclusion from Figure 3. Lastly, while uDSR — an ensemble method — still achieves higher overall performance than DDSR, it is important to note that DDSR can be seamlessly incorporated into the uDSR pipeline. It could either replace DSR or be added as a complementary component, further enhancing the ensemble's effectiveness in symbolic expression discovery.

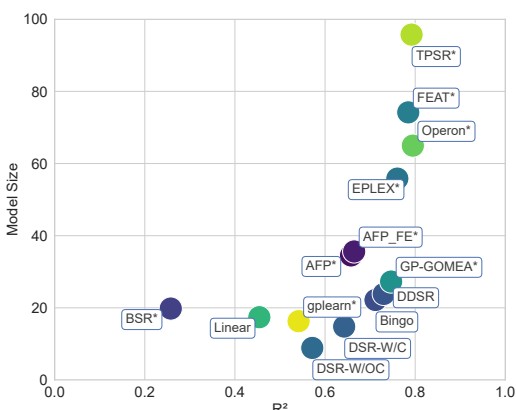

Figure 3: Mean $R^2$ score *vs.* model size on black-box problems. Model size means the expression length. The numerical values are reported in Appendix Table 7.

For the black-box problems — where ground-truth solutions are unavailable — we evaluated model performance using the trade-off between average $R^2$ score and expression size, as shown in Figure 3. DDSR lies at the frontier of the Pareto curve, indicating that it achieves one of the best balances between accuracy and interpretability. In other words, DDSR offers high data-fitting performance while maintaining relatively concise expressions. As an example for comparison, TPSR achieves slightly higher $R^2$ scores, but the resulting expressions are substantially more complex, averaging around 70 tokens.

**Running Time.** Average run-time of each method is reported in Appendix Table 6. When using the same token space, DDSR requires only about 1/2 of the runtime of DSR (specifically, DSR-W/C), highlighting the training efficiency enabled by our diffusion-based framework. This might be attributed to our diffusion-based model, which generates more diverse and high-quality candidates, thereby accelerating expression discovery. Note that DSR-W/OC took less running time due to its reduced token space — no constant tokens — and the thereof lack of constant optimization step.

Table 2: Performance of DSR w/wo Diffusion.

|  | DSR | Diffusion-DSR | DDSR |
|---|---|---|---|
| Solution Rate (%) | 24.8 | 44.6 | 46.5 |
| Accuracy Rate (%) | 38 | 57 | 60 |

### 5.2 Ablation Studies

**Advantage of Diffusion.** To isolate the effect of diffusion, we retain only the diffusion component and keep all other parts of our SR model identical to DSR — namely, the risk-seeking policy gradient (w/o GPRO) and short-term risk seeking. We refer to this variant as *Diffusion-DSR*. The symbolic solution rate and accuracy rate on SRBench (0% noise) are reported in Table 2. Results show that diffusion alone substantially improves DSR, and adding our other components yields further gains. This demonstrates that diffusion is a key driver of the overall improvement.

Table 3: Average number of novel expressions (diversity) generated across Feynman problems.

| Epoch | DDSR | DSR | Improvement (%) |
|---|---|---|---|
| 0 | 951.8 | 989.8 | -3.8 |
| 50 | **917.3** | 804.1 | 14.1 |
| 150 | **928.0** | 720.7 | 28.8 |
| 200 | **914.0** | 710.2 | 28.7 |
| 250 | **869.7** | 655.3 | 32.7 |
| 300 | **812.8** | 656.1 | 23.9 |
| 350 | **756.0** | 578.1 | 30.8 |
| 400 | **710.6** | 558.5 | 27.2 |
| 450 | **684.5** | 536.0 | 27.7 |
| 500 | **662.0** | 468.0 | 41.5 |

**Expression Diversity.** We examined the average number of novel expressions generated during training across all Feynman problems. The results, shown in Table 3 and Appendix Figure 9, indicate that DDSR consistently discovers more novel expressions than DSR during training, with an average diversity improvement of about 28% and a peak improvement of 41%. These findings confirm that DDSR effectively explores a broader range of expressions, thereby facilitating expression discovery.

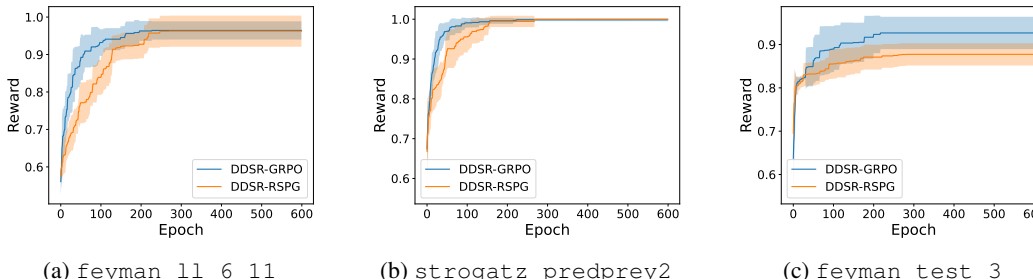

(a) `feyman_ll_6_11`    (b) `strogatz_predprey2`    (c) `feyman_test_3`

Figure 5: Learning curves for DDSR with GPRO and with RSPG. Error bars denote one standard deviation.

**Individual Components.** Next, we conducted ablation studies to assess the effectiveness of individual components in our method. Specifically, we evaluated three variants of DDSR by: (1) replacing our random mask-based discrete diffusion with the standard Discrete Denoising Diffusion Probabilistic Model (D3PM) (see Section 2); (2) replacing Grouped Relative Policy Optimization (GRPO) with the standard risk-seeking policy gradient (RSPG); (3) substituting our Long-Short-Term (LST) policy with a short-term (ST) policy that selects the top $\alpha\%$ expressions only from the current model. We tested these variants on the Feynman and Strogatz datasets in SRBench. The solution accuracy results are shown in Figure 4.

First, our random mask-based diffusion improves solution accuracy rate — by 2.2% on the Feynman dataset and 35.7% on the Strogatz dataset — compared to the standard D3PM. This improvement may stem from the fact that D3PM perturbs all tokens at each diffusion step, which can introduce instability during training. In contrast, our method perturbs only one token at each step, providing more stable and effective learning dynamics.

Second, while RSPG leads to a modest 0.4% improvement on the Feynman dataset, it results in a 3.5% performance drop on the Strogatz dataset relative to GRPO. This highlights GRPO's robustness. Moreover, GRPO accelerates training:

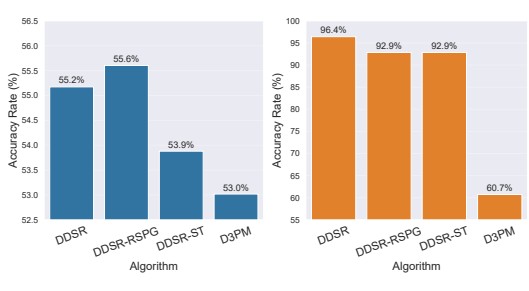

(a) Feynman dataset    (b) Strogatz dataset

Figure 4: DDSR ablations accuracy rate (%) for the Feynman and Strogatz datasets.

on average, it converges 30 epochs faster than RSPG. Representative learning curves for randomly selected problems are shown in Figure 5 and Appendix Figure 8.

Third, the LST policy outperforms the ST policy by 1.3% and 3.5% on the Feynman and Strogatz datasets, respectively. This demonstrates the benefit of leveraging historically top-performing expressions to enhance exploitation.

**Choice of $\alpha$.** Our main results are reported with $\alpha = 5\%$, following the same setting as DSR — our primary competing baseline — for a fair and strong comparison. We further examine how different choices of $\alpha$ affect the trade-off between computational cost and performance. The details are provided in Appendix D.1.

# 6    Conclusion

We have introduced DDSR— a random masked discrete diffusion model for symbolic regression. Our experimental results on the SRBench benchmark demonstrate that DDSR outperforms deep reinforcement learning-based approaches and achieves performance comparable to state-of-the-art genetic programming (GP) methods.

However, DDSR has some limitations. In particular, it tends to require longer runtimes to discover solutions for complex problems and exhibits reduced robustness to high levels of noise in the data.

Future work may explore extending the discrete diffusion framework to supervised learning of foundational models, improving the efficiency of the diffusion transformation process, and refining the reward function to enhance exploitation and increase robustness to noisy data.

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

# Appendix

## A  Algorithms

### A.1  Sampling Expressions

We sample an expression from a distribution matrix of size $M \times d$, where $M$ denotes the maximum length of the expression, and $d$ denotes the number of tokens in the library. At each step in the process, we get a vector of valid tokens for the current node based on the current incomplete version of $\tau$. This vector enforces rules to avoid the expression from being invalid. DDSR further restricts that the sampled expression trees can include up to 10 constants, and trigonometric functions such as $\sin$ and $\cos$ can not be nested. The rule for setting a maximum number of constant tokens can help reduce optimization runtime and mitigate overfitting. Constant tokens are optimized with the Levenberg-Marquardt algorithm (Levenberg, 1944) for each discovered equation. The prevention of nested trigonometric functions is inherited from DSR, as the authors of DSR claimed nested trigonometric functions do not occur in physics.

---

**Algorithm 2** Sampling expressions for a given sequence of categorical distributions

---

**input** probability distribution $p$, max depth of the tree $M$
**output** An expression $\tau$
 1: **for** $i = 0$ to $M$ **do**
 2:    $r \leftarrow$ Get Valid Tokens$(\tau, i)$
 3:    $p_i \leftarrow p_i \cdot r$
 4:    **if** $\sum_{j=1}^{d} p_{i,j} = 0$ **then**
 5:        $p_i \leftarrow \mathbf{1} * r$
 6:    **end if**
 7:    $p_i \leftarrow p_i / \sum_{j=1}^{d} p_{i,j}$
 8:    $\tau_i \sim p_i$
 9: **end for**
10: **return** $\tau$

---

**Ordering:**  Converting expression trees into a vector of tokens can be done in various ways. We used the tree's breadth-first search ordering (BFS) to order the nodes/tokens. This ordering keeps siblings of parents nodes close to each other in the ordering. In comparison, the preorder traversal ordering (POT) can place siblings far apart. Figure 6 shows the conversion of an expression tree into a breadth-first search ordering and the comparison to a preorder traversal ordering.

## B  Model Architecture

The positional encoding in the input layer enables the attention layers to capture positional-based relationships of the tokens. In addition to generating tokens from the BFS search order, our diffusion model also involves a time step variable $t$ informing the progress of the diffusion and denoising. To integrate the information from both token positions $l$ and time steps $t$, we introduce a two-dimensional encoding as defined below. The full architecture of our model is shown in Figure 7. The hyperparameter settings are listed in Table 4.

$$\mathrm{PE}(l,t)_{2i} = \sin(\frac{l}{10000^{(4i/D)}}), \mathrm{PE}(l,t)_{2i+1} = \cos(\frac{l}{10000^{(4i/D)}}),$$
$$\mathrm{PE}(l,t)_{D+2j} = \sin(\frac{t}{10000^{(4j/D)}}), \mathrm{PE}(l,t)_{D+2j+1} = \cos(\frac{t}{10000^{(4j/D)}}), \tag{7}$$

## C  Noise Setting in SRBench

The noise levels indicate the relative amount of noise added to the expression values for evaluation purposes. Specifically, for each data point, Gaussian noise is added as $y = f(x_1, \ldots, x_k) +$ noise-level $\cdot s \cdot \epsilon$, where $f(x_1, \ldots, x_k)$ is the ground-truth expression, $s$ is the standard deviation of

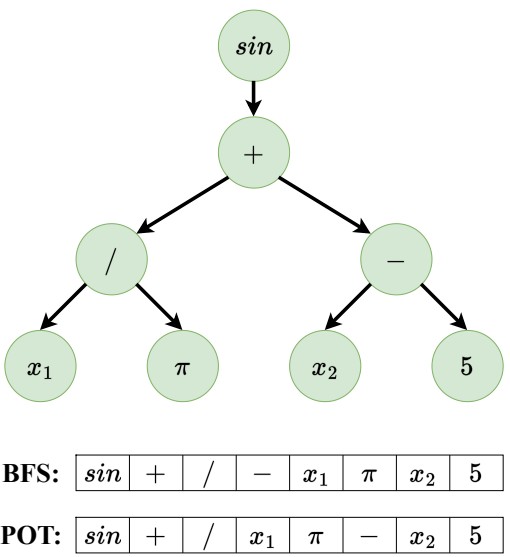



Figure 6: Breadth first search ordering of an expression compared to the pre-order traversal

Table 4: Hyperparameter settings of DDSR.

| Hyperparameter | DDSR |
|---|---|
| Variables | $\{1, c \text{ (Constant Token)}, x_i\}$ |
| Unary Functions | $\{\sin, \cos, \log, \sqrt{(\cdot)}, \exp\}$ |
| Binary Functions | $\{+, -, *, /, \hat{\ }\}$ |
| Batch Size | 1000 |
| Risk Seeking Percent ($\alpha$) | 5% |
| Optimizer | ADAM |
| Learning Rate | 1E-4 |
| Max Depth | 32 |
| Oversampling | 3 |
| Number of Epochs | 600 |
| Entropy Coefficient $\lambda$ | 0.0005 |
| Encoder Number | 1 |
| Decoder Number | 1 |
| Number of Heads | 1 |
| Feed Forward Layers Size | 2048 |
| $\beta$ | 0.01 |
| $\epsilon$ | 0.2 |
| Embedding Dim | 15 |
| $C$ | 5 |
| $G$ | 5 |

the expression values across all inputs $(x_1, \ldots, x_k)$ in the dataset, and $\epsilon$ is drawn from a standard Gaussian white noise distribution $\mathcal{N}(0, 1)$. This setup is the standard evaluation protocol in the SRBench benchmark, which is designed to assess the robustness of symbolic regression methods under varying noise levels.

## D   Additional Results

Table 5 provides the numerical values of the symbolic solution rate, accuracy rate and simplified complexity of all the methods offered by SRBench for problems with known solutions. Table 8 reports the $R^2$ scores, model size, and training time of every method on the black-box problems of SRBench. DDSR has similar performance to Bingo and GP-GOMEA on the black box problems, performing as an improvement in reducing the complexity of GP-GOMEA and increasing the $R^2$

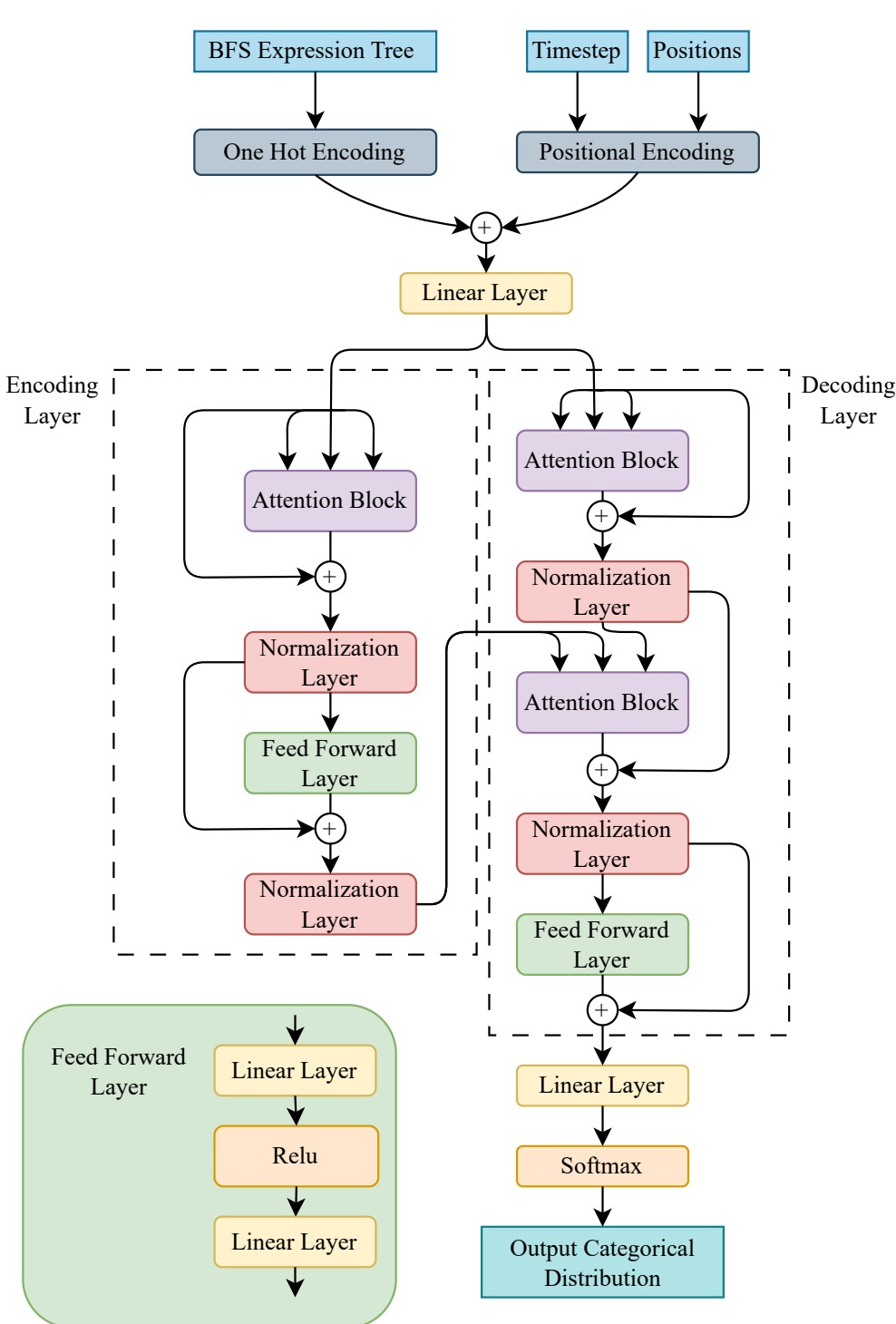

Figure 7: The architecture of the diffusion model in DDSR.

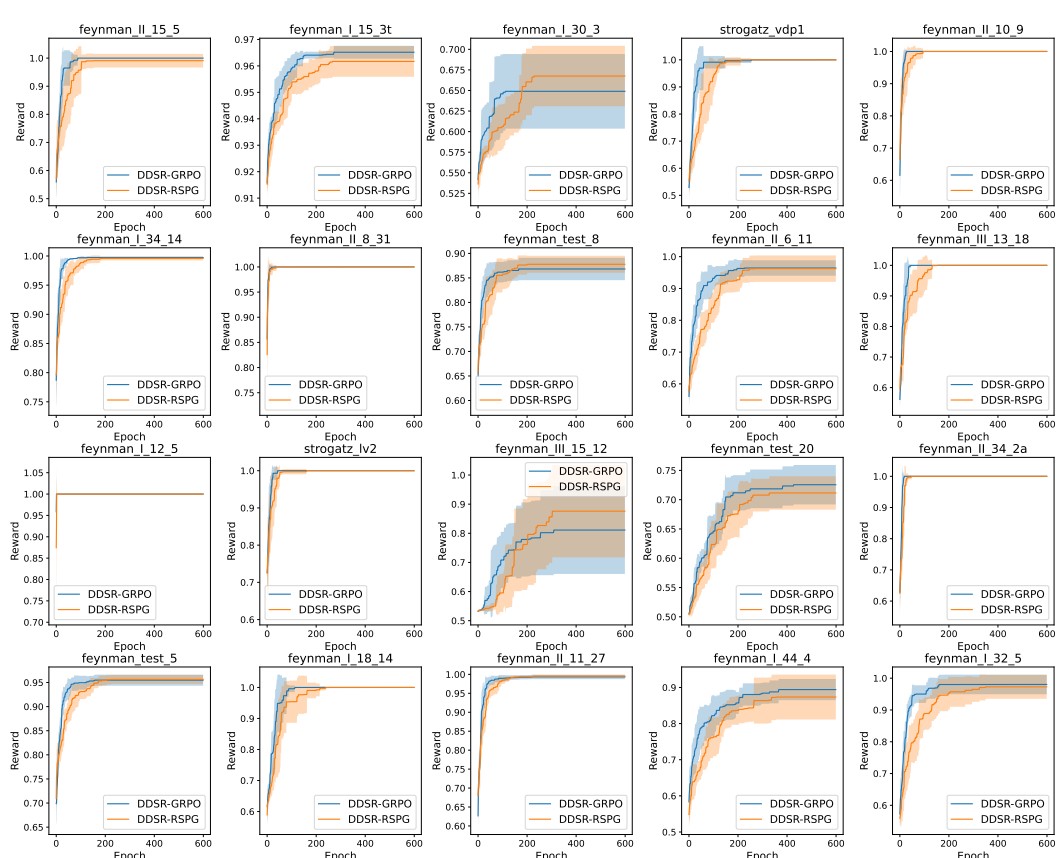

Figure 8: Learning curves of 20 problems for DDSR with GPRO and with RSPG. These problems are randomly selected from the Feynman and Strogatz dataset. Colored regions denote one standard deviation.

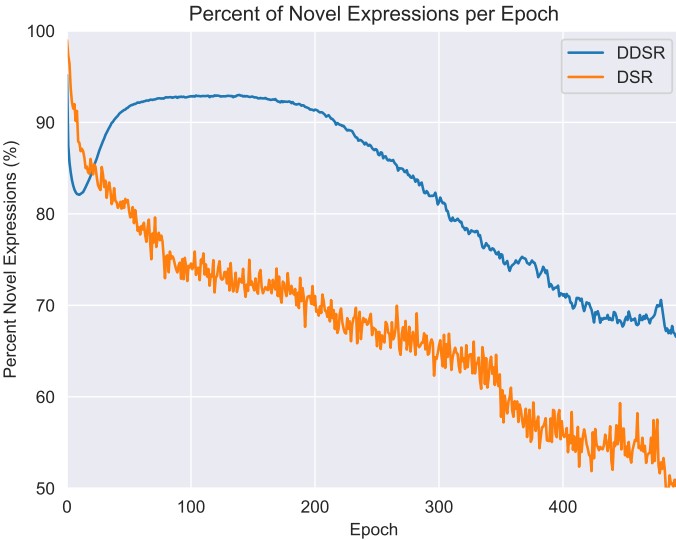

Figure 9: Comparing expression novelty throughout training for DDSR and DSR

score of Bingo. Significantly, DDSR outperforms four of the five machine learning methods from SRBench (linear fit, random forests, AdaBoost, and MLPs) in $R^2$ while offering an interpretable model. XGBoost is the only machine learning method that outperforms DDSR in $R^2$ score by $0.046$ while increasing the model size by 713x and losing any natural interpretability.

Table 5: Performance on SRBench problems with known solutions.

| Algorithm | Symbolic Solution Rate (%) | | | | Accuracy Rate (%) | | | | Simplified Complexity | | | |
|---|---|---|---|---|---|---|---|---|---|---|---|---|
| | 0.0 | 0.001 | 0.01 | 0.1 | 0.0 | 0.001 | 0.01 | 0.1 | 0.0 | 0.001 | 0.01 | 0.1 |
| AFP | 22.31 | 19.92 | 16.85 | 12.85 | 43 | 42 | 40 | 41 | 29.35 | 28.43 | 29.01 | 30.82 |
| AFP_FE | 28.08 | 22.69 | 20.31 | 12.85 | 56 | 50 | 50 | 50 | 34.58 | 35.66 | 34.45 | 36.77 |
| AIFeynman | 61.84 | 31.89 | 12.61 | 0.86 | 74 | 74 | 68 | 10 | 83.29 | 88.66 | 99.27 | 110.54 |
| BSR | 2.50 | 0.61 | 0.08 | 0.00 | 12 | 11 | 12 | 7 | 34.29 | 35.51 | 36.80 | 38.38 |
| Bingo | 48.77 | 14.62 | 4.77 | 0.77 | 64 | 60 | 62 | 59 | 15.56 | 19.29 | 21.32 | 22.54 |
| DDSR | 46.54 | 27.02 | 20.33 | 10.69 | 60 | 60 | 59 | 56 | 17.33 | 17.13 | 17.87 | 18.48 |
| DSR-W/C | 24.81 | 24.42 | 17.53 | 10.48 | 38 | 41 | 40 | 39 | 16.57 | 16.10 | 16.96 | 18.45 |
| DSR-W/OC | 19.71 | 19.23 | 18.92 | 16.61 | 24 | 25 | 25 | 25 | 13.14 | 14.36 | 14.61 | 14.40 |
| EPLEX | 12.50 | 9.92 | 8.77 | 9.54 | 44 | 45 | 52 | 47 | 53.24 | 51.74 | 49.91 | 40.04 |
| FEAT | 0.10 | 0.00 | 0.00 | 0.00 | 40 | 43 | 41 | 14 | 88.01 | 77.32 | 72.61 | 50.40 |
| FFX | 0.00 | 0.00 | 0.00 | 0.08 | 0 | 0 | 3 | 18 | 274.88 | 273.29 | 286.03 | 341.38 |
| GP-GOMEA | 43.08 | 10.62 | 4.69 | 1.46 | 71 | 70 | 73 | 68 | 25.73 | 32.75 | 37.59 | 45.41 |
| ITEA | 20.77 | 13.77 | 7.69 | 1.46 | 27 | 27 | 27 | 26 | 14.46 | 14.96 | 15.35 | 16.00 |
| MRGP | 0.00 | 0.00 | 0.00 | 0.00 | 93 | 92 | 89 | 2 | 109.95 | 106.50 | 83.06 | 0.00 |
| Operon | 16.00 | 12.31 | 1.92 | 0.08 | 87 | 86 | 86 | 73 | 40.80 | 40.13 | 60.40 | 70.78 |
| SBP-GP | 22.69 | 0.69 | 0.00 | 0.00 | 74 | 74 | 75 | 54 | 109.94 | 102.09 | 112.93 | 116.30 |
| TPSR | 36.09 | 0.00 | 0.00 | 0.00 | 68 | 68 | 66 | 38 | 57.11 | 59.32 | 63.42 | 65.83 |
| gplearn | 16.15 | 16.86 | 16.59 | 16.00 | 30 | 29 | 27 | 22 | 45.80 | 37.76 | 36.42 | 33.84 |

Table 6: Average run time for each method on SRBench symbolic dataset.

| Algorithm | Run Time (s) |
|---|---|
| AFP | 3488.63 |
| AFP_FE | 28830.37 |
| AIFeynman | 29590.66 |
| Bingo | 22542.44 |
| BSR | 28800.16 |
| DDSR | 14441.12 |
| DSR-W/C | 27131.73 |
| DSR-W/OC | 630.83 |
| EPLEX | 10865.90 |
| FEAT | 1079.46 |
| FFX | 17.34 |
| GP-GOMEA | 2072.25 |
| ITEA | 1411.92 |
| MRGP | 18527.59 |
| Operon | 2483.09 |
| SBP-GP | 28968.48 |
| TPSR | 172.87 |
| gplearn | 1396.98 |

## D.1 Ablation Study on Risk-Seeking Percentage $\alpha$

For a fair and strong comparison, we set $\alpha = 5\%$, following the same setting as DSR (Petersen et al., 2019), which is our primary competing baseline. To investigate the trade-off between the memory cost for back-propagation at each training step and solution discovery performance, we varied $\alpha$ from $1\%$, $5\%$, $10\%$ and $25\%$, and ran our method on the Strogatz dataset. The GPU memory usage and the solution accuracy rate are reported in Table 8. We can see the best trade-off is achieved at $5\%$.

Table 7: Performance on black-box problems of SRBench.

| Algorithm | $R^2$ Score | Complexity |
|---|---|---|
| AFP | 0.657613 | 34.5 |
| AFP_FE | 0.664599 | 35.6 |
| AIFeynman | -3.745132 | 2500 |
| AdaBoost | 0.704752 | 10000 |
| BSR | 0.257598 | 19.8 |
| Bingo | 0.711951 | 22.2 |
| DDSR | 0.730218 | 23.8 |
| DSR-W/OC | 0.571669 | 8.89 |
| DSR-W/C | 0.642417 | 14.8 |
| EPLEX | 0.760414 | 55.8 |
| FEAT | 0.784662 | 74.2 |
| FFX | -0.667716 | 1570 |
| GP-GOMEA | 0.746634 | 27.3 |
| ITEA | 0.640731 | 112 |
| KernelRidge | 0.615147 | 1820 |
| LGBM | 0.637670 | 5500 |
| Linear | 0.454174 | 17.4 |
| MLP | 0.531249 | 3880 |
| MRGP | 0.417864 | 12100 |
| Operon | 0.794831 | 65.0 |
| RandomForest | 0.698541 | 1.54e+06 |
| SBP-GP | 0.798932 | 639 |
| TPSR | 0.792001 | 95.7 |
| XGB | 0.775793 | 16400 |
| gplearn | 0.541264 | 16.3 |

Table 8: DDSR performance on Strogatz dataset with different choices of $\alpha$.

| $\alpha$ | 1% | 5% | 10% | 25% |
|---|---|---|---|---|
| Accuracy Rate | 89% | 96% | 92% | 92% |
| GPU Memory (MB) | 220 | 270 | 348 | 530 |

