# OpenReview forum: "Diffusion-Based Symbolic Regression"
_ICLR.cc/2026/Conference — Submitted to ICLR 2026_

### Official Review · Reviewer_nKhr · 2025-10-27

**Soundness:** 3
**Presentation:** 2
**Contribution:** 3
**Rating:** 4
**Confidence:** 3

**Summary:**

This paper introduces Diffusion-Based Symbolic Regression (DDSR), a novel approach that frames symbolic regression (SR) as a generative modeling task using a discrete diffusion model. The core of the method is a "random mask-based" diffusion process, where equations (represented as token sequences) are gradually corrupted by masking one token at a time. A Transformer-based model is trained to reverse this process, learning to denoise a fully masked sequence into a valid, high-quality equation.

The model is trained in an unsupervised manner using reinforcement learning (RL) on the measurement dataset. The authors integrate the diffusion generator with two key components:

Token-wise Group Relative Policy Optimization (GRPO): An efficient and stable policy gradient method to optimize the RL objective.

Long Short-Term Risk-Seeking: An experience replay-like buffer that maintains a pool of the best-performing equations discovered across all training epochs, preventing the model from "forgetting" good candidates.

Experiments on the SRBench benchmark show that DDSR significantly outperforms Deep Symbolic Regression (DSR), a major RL-based baseline, in both solution rate and accuracy. Furthermore, DDSR is competitive with strong Genetic Programming (GP) methods while generating substantially simpler and more interpretable expressions.

**Strengths:**

Novelty and Conceptual Framing: The paper introduces a new and elegant generative framework (discrete diffusion) to symbolic regression. The hypothesis that diffusion's strength in generating diverse samples would help explore the expression space and avoid mode collapse (a common problem in SR) is well-motivated and convincingly proven.

Performance Gains: DDSR substantially outperforms its most direct and important baseline, Deep Symbolic Regression (DSR). On the key 0% noise benchmark, it more than doubles the symbolic solution rate of the comparable DSR-W/C variant (46.5% vs. 24.8%).

Interpretability-Accuracy Trade-off: While other methods (like GP-GOMEA or TPSR) may achieve high accuracy, they often do so with very complex, uninterpretable expressions. DDSR finds expressions that are dramatically simpler (e.g., avg. complexity 17.7 vs. 61.4 for TPSR) while remaining highly accurate and on the Pareto frontier (Figure 3).

**Weaknesses:**

Contradictory Runtime Claim: the paper's primary weakness is a glaring contradiction. The experimental results (Section 5.1) claim DDSR is twice as fast as DSR-W/C. The conclusion (Section 6) claims a limitation is "longer runtimes for complex problems". This must be clarified. Does DDSR have a lower average runtime but a higher worst-case runtime? This confusion needs to be resolved.

Questionable Novelty and Misleading "Diffusion" Framing: The paper's central claim to novelty rests on applying a "diffusion" model. However, the described method, masking one token at a time and having T=M steps (where M=32), bears little resemblance to the standard DDPM/D3PM framework. It appears to be a standard masked language model trained with RL

Contradictory Ablation Study: The ablation in Figure 4 is intended to justify the GRPO and LST components, but it is confusing and self-contradictory. For the Feynman dataset (Fig 4a), the baseline DDSR-RSPG (55.6%) performs better than the full DDSR model (55.2%), which uses the proposed GRPO. This result directly invalidates the claim that GRPO is a necessary or beneficial component for this task.

**Questions:**

Runtime Contradiction: Could you please clarify the discrepancy regarding runtime? Section 5.1 states DDSR is ∼2× faster than DSR-W/C, but the Conclusion lists "longer runtimes for complex problems" as a limitation. Does this mean DDSR has a lower average runtime but a higher variance or worse worst-case runtime on the most difficult problems?

GRPO Ablation Contradiction: In Figure 4(a), DDSR-RSPG (55.6%) achieves a higher accuracy rate than the full DDSR (55.2%). This suggests that your proposed GRPO component is hurting performance on the Feynman dataset. How do you reconcile this with your claim that GRPO improves performance?

Noise Robustness: The method performs worse than the DSR-W/OC baseline on 10% noise. Why does the proposed diffusion framework fail so significantly when noise is introduced, and how does this limit the practical applicability of the method?

Novelty vs. MLM: How is the proposed method (masking one token at a time over M steps) technically different from a standard Masked Language Model (like BERT) being trained to generatively fill masks, combined with an RL objective? Why is this framed as "diffusion," a term that typically implies a much longer, gradual noising process?

---

### Official Review · Reviewer_Qyo2 · 2025-10-31

**Soundness:** 2
**Presentation:** 3
**Contribution:** 3
**Rating:** 4
**Confidence:** 3

**Summary:**

The paper has proposed a diffusion-based method for symbolic regression, integrating GRPO and long short-term risk seeking policy and achieving competitive performance comparing to other methods. The method is effective based on experiment results, the weaknesses still need to be addressed.

**Strengths:**

1. The authors proposed a novel method for symbolic regression that combines diffusion methods and reinforcement learning, providing supportive evaluation against 18 baselines.
2. The method can generate free-form expression with much less tokens comparing to other methods, which is promising.
3. The paper is well-writing, with systematical explanations of experiments via numeric results and figures.

**Weaknesses:**

1. Not enough datasets for comprehensive evaluation. In the paper, the authors only use 1 dataset that has 133 problems with ground-truth and 120 problems without solutions, which is relatively small size relative to diffusion models with transformer-based architectures and might incur overfitting problem. The difference among SRbench, Feynman and Strogatz datasets should be stated clearly.
2. The ablation study replaces some key components with other components from prior literature, whereas this can only show the model has better performance against former ones and the contribution of each component for current model is still not clear.
3. The authors mentioned a key difference with former methods: masking only 1 token per step, which avoid abrupt distortions and smoothly blur the expression structure. For this point, an experiment of number of token to mask each time need to be conducted to support the statement, and how this can reduce the denoising steps/computational cost.
4. Readability for figures should be improved. For fig 2, the confidence interval tangles and it’s hard to distinguish, so better use different colors. Fig 3 should also use more distinctive colors. Highlighting the proposed method would be beneficial.
5. Misleading terms. When analyzing table 1, as you have solution rate and accuracy rate, it is better not say solution accuracy in “However, even at the 10% noise … than DSR-W/OC”. For fig 3, better use expression length instead of model size.

**Questions:**

1. The author mentioned that DDSR generates more diverse expressions over DSR in tab 3, what is the measurement of the diversity in this case? Will DDSR generate more diverse but less symbolic correct/meaningful/reasonable expressions (like f(x)=x/0) comparing to DSR?
2. The long short-term risk-seeking policy maintains a pool of top candidates from all epochs. Could the removal of the bottom a% keep the pool size fixed or the size changes every step? How the size of pool affects compute speed?
3. While the quantitative results show the advantages of the method, it would be better if examples of generated expression can be shown for intuition.

---

### Official Review · Reviewer_h5dJ · 2025-11-01

**Soundness:** 3
**Presentation:** 3
**Contribution:** 1
**Rating:** 2
**Confidence:** 5

**Summary:**

This paper suggests to use masked diffusion framework on the symbolic regression task, with applying GRPO reinforcement learning.

**Strengths:**

The paper structure is clear and easy to follow. The experiments are extensive.

**Weaknesses:**

- Novelty: I fail to see the novelty of this work. The authors claim to apply masked diffusion and GRPO to the symbolic regression task; however, both are well-established frameworks with clear prior formulations. The paper does not highlight any critical adaptations or theoretical insights specific to symbolic regression that would justify a new contribution. Without such task-specific modifications, the method essentially reduces to a straightforward application of existing components.
- Long Short-Term Risk-Seeking: the authors claim to expand the candidate pool to include not only the top-performing samples from the current epoch but also those from all previous epochs. However, this seems to be a minor heuristic rather than a good improvement, and no in-depth analysis is provided.
- Theorem: Following the above, for the expanding of candidate pool, more importantly, if we revisit the derivation of GRPO, it is based on the basic intuition that maximizing the expected reward under the current policy (sample from current policy). GRPO suggests the importance ratio $\pi_{\theta}$ / $\pi_{old}$, so that the reward is under the old policy (usually the last epoch, so we sample from last epoch). Extending the candidate set to include samples from all previous epochs would require modifying this formula, e.g., adding all previous policy in the importance ratio. Otherwise, the theoretical justification of the GRPO objective no longer holds. While such a modification may work empirically, the paper does not discuss any adaptation to maintain the validity of the optimization objective.

Overall, the main weakness of this paper is the lack of methodological innovation, and the work mainly applies existing frameworks (masked diffusion and GRPO) to symbolic regression without introducing task-specific adaptations.

**Questions:**

The paper would benefit from clarifying what new adaptations are specifically introduced for symbolic regression, beyond combining masked diffusion and GRPO. Please highlight which components are unique or necessary, and provide deeper analysis to show how these adaptations contribute to performance.

---

### Official Review · Reviewer_ESxh · 2025-11-07

**Soundness:** 3
**Presentation:** 2
**Contribution:** 3
**Rating:** 6
**Confidence:** 3

**Summary:**

This paper proposes DDSR, a diffusion-based approach for symbolic regression, inspired by DSR. DDSR employs a random mask-based diffusion based approach in place of the RNN used in DSR. In addition, the authors introduce improvements to the RL training setup, such as a token-wise GRPO, and a Long Short-Term risk seeking policy. Experimental results show that DDSR outperforms DSR and several other baselines in symbolic recovery rate, accuracy and simplicity of generated expressions. In general, DDSR appears to strike a good balance between accuracy and simplicity.

**Strengths:**

1. The paper is generally well-written and clear in its contributions.

2. The topic, finding interpretable analytical relationships, is important and relevant to ICLR.

3. The idea is fairly novel: as far as I know, the combination of Diffusion Models and Symbolic Regression hasn’t been explored before. DDSR seems like a solid and interesting improvement over DSR.

5. There is good empirical validation. Results on the SRBench show that DDSR outperforms the original DSR in symbolic solution rate, accuracy, and matches its simplified complexity. There appears to be a good trade-off between accuracy and simpler expressions.

**Weaknesses:**

1. This isn’t exactly a weakness, but I have doubts whether this model can actually be called a diffusion model. The “diffusion” process here masks one token at a time instead of adding noise to all tokens, and there is no Markov chain. Mathematically it seems different, and it would perhaps be better to call it a diffusion-inspired model. Please correct me if I’ve misunderstood this.

2. The discussion around the 10% noise case is somewhat confusing. The concept of “token space” isn’t previously defined or explained in a sufficient manner. Could the authors clarify this topic?

3. The paper claims DDSR produces more interpretable symbolic expressions. Showing examples of a few cases would strengthen this claim.

4. Figure 4 should use a consistent y-axis scale across panels; right now it can be misleading. Also, some tables (such as Table 5) could be improved, by bolding the best results for example.

5. The runtime comparison should have more emphasis, as it is actually very important. Table 6 could be moved to the main paper and it would help to see plots that compare accuracy/symbolic solution rate/simplicity and runtime.

6. The results show that DDSR performs worse in pure accuracy when compared to some GP methods. It would be interesting to analyze where this happens and provide a more comprehensive discussion on these cases.

**Questions:**

1. In Figure 3, is the axis label referring to model size or expression size? There seems to be a mismatch with the text.

2. Also in Figure 3, why is uDSR missing?

3. Could Table 2 (the ablation) be repeated for different noise levels to offer a more comprehensive view?

---

### Meta-Review · Area_Chair_L9aR · 2026-01-07

**Summary:**

The paper proposes DDSR, a diffusion-inspired generative reinforcement learning framework for symbolic regression.

Reviewers agree the problem is important and the method shows empirical improvements over DSR with simpler expressions.

However,  the technical novelty is unclear, the diffusion formulation is questionable, and the methodological justification and evaluation are insufficient.

The authors did not provide a rebuttal to address the reviewers’ concerns.

Overall, the paper does not meet the acceptance bar.

**Reviewer Concerns:**

No rebuttals for reviewers' concerns.

**Reviewer Scores:**

No rebuttals and changes.

---

### Decision · Program_Chairs · 2026-01-26

Reject